# Transferability of Published Population Pharmacokinetic Models for Apixaban and Rivaroxaban to Subjects with Obesity Treated for Venous Thromboembolism: A Systematic Review and External Evaluations

**DOI:** 10.3390/pharmaceutics15020665

**Published:** 2023-02-16

**Authors:** Cyril Leven, Pauline Ménard, Isabelle Gouin-Thibault, Alice Ballerie, Karine Lacut, Edouard Ollier, Jérémie Théreaux

**Affiliations:** 1Inserm, UMR 1304 (GETBO), Western Brittany Thrombosis Study Group, Western Brittany University, 29238 Brest, France; 2Department of Biochemistry and Pharmaco-Toxicology, Brest University Hospital, 29200 Brest, France; 3Department of Hematology, Brest University Hospital, 29200 Brest, France; 4Hematology Laboratory, Rennes University Hospital, 35000 Rennes, France; 5Inserm, EHESP, Irset (Institut de Recherche en Santé, Environnement et Travail)-UMR_S 1085, Université de Rennes, 35708 Rennes, France; 6Department of Internal Medicine and Clinical Immunology, Rennes University Hospital, 35000 Rennes, France; 7Internal Medicine, Vascular Medicine and Pneumology Department, Brest University Hospital, 29200 Brest, France; 8SAINBIOSE, INSERM U1059, Université de Lyon, 69621 Saint-Etienne, France; 9Department of General, Digestive and Metabolic Surgery, Brest University Hospital, 29200 Brest, France

**Keywords:** apixaban, rivaroxaban, obesity, venous thromboembolism, external validation, pharmacokinetics

## Abstract

Apixaban and rivaroxaban have first-line use for many patients needing anticoagulation for venous thromboembolism (VTE). The pharmacokinetics of these drugs in non-obese subjects have been extensively studied, and, while changes in pharmacokinetics have been documented in obese patients, data remain scarce for these anticoagulants. The aim of this study was to perform an external validation of published population pharmacokinetic (PPK) models of apixaban and rivaroxaban in a cohort of obese patients with VTE. A literature search was conducted in the PubMed/MEDLINE, Scopus, and Embase databases following the PRISMA statement. External validation was performed using MonolixSuite software, using prediction-based and simulation-based diagnostics. An external validation dataset from the university hospitals of Brest and Rennes, France, included 116 apixaban pharmacokinetic samples from 69 patients and 121 rivaroxaban samples from 81 patients. Five PPK models of apixaban and 16 models of rivaroxaban were included, according to the inclusion criteria of the study. Two of the apixaban PPK models presented acceptable performances, whereas no rivaroxaban PPK model did. This study identified two published models of apixaban applicable to apixaban in obese patients with VTE. However, none of the rivaroxaban models evaluated were applicable. Dedicated studies appear necessary to elucidate rivaroxaban pharmacokinetics in this population.

## 1. Introduction

Overweight is a global health problem, and its worldwide prevalence is steadily increasing. In Europe, the prevalence of obesity, defined by a body mass index (BMI) ≥ 30 kg/m², has reached 20% in the most affected countries [1]. In the United States, the prevalence estimates were close to 40% of the adult population [2], and the recent data on children and adolescents suggested an acceleration of this trend during the COVID-19 pandemic [3]. Among other conditions, obesity is associated with an increased risk of cardiovascular morbidity and mortality, including venous thromboembolism (VTE) and a higher risk of recurrent VTE following the withdrawal of anticoagulation therapy [4]. Direct-acting oral anticoagulants (DOACs) have first-line use for many patients needing anticoagulation for VTE [5]. Compared to anti-vitamin K drugs, DOACs have a favorable benefit-risk profile with fewer drug interactions and a lower incidence of intracranial hemorrhage [6]. Their ease of use due to fixed-dose regimens and lack of routine monitoring makes DOACs an attractive therapeutic option for the management of VTE in people with high weight or BMI. 

Although the PK of apixaban and rivaroxaban in non-obese subjects has been extensively studied [7,8], data on obese patients remain scarce. Yet, substantial changes in pharmacokinetics (PK) are observed in persons with obesity [9,10,11]; these effects are not consistent across drugs, and they have not been clearly characterized for DOACs. The two PK parameters most likely to be altered in this population are the volume of distribution, and drug elimination [9,11]. Increases in the volume of distribution are particularly seen for lipophilic drugs and are usually not proportional to body weight [10]. The impact of obesity on drug elimination clearance also depends on several drug- and patient-related factors, such as the cytochromes involved, variations in hepatic blood flow, and the duration of obesity. Therefore, it remains difficult to assess the impact of obesity on drug elimination. Although lean body weight is increasingly proposed as a measure of body mass in obesity pharmacology, particularly for renally eliminated drugs [12], there is still no consensus on the most appropriate descriptor of body weight in this population.

The initial recommendations published by the ISTH in 2016 for the use of DOACs in patients with obesity for the treatment and prevention of VTE were conservative [13]. The guidance suggested not using DOACs in patients with extreme obesity (BMI > 40 kg/m² or weight > 120 kg), and, if DOACs were nevertheless used in these patients, to monitor the peak and trough drug levels [13]. These recommendations were made in light of insufficient clinical evidence regarding efficacy and safety in patients with extreme obesity, as phase 3 clinical trials comparing DOACs with warfarin for the treatment of VTE included few patients with obesity and extreme obesity. Even though prospective and specific studies in this population are still lacking, especially in patients with morbid obesity, the recent update of these guidelines presented standard doses of rivaroxaban or apixaban as appropriate options for the treatment of deep vein thrombosis, regardless of patient weight or high BMI [14]. The monitoring of DOAC levels was no longer recommended because there were insufficient data to influence management decisions.

While there are no warning signs regarding the efficacy or safety of apixaban or rivaroxaban in patients with obesity, the ISTH guidelines are recent and have many limitations, and the question of the impact of obesity on the PK of DOACs remains unanswered. Several population PK (PPK) models were developed to characterize the complete PK of apixaban and rivaroxaban. A PPK analysis can identify and explain the determinants of inter-individual variability in drug exposure, and it is frequently used to guide drug development and inform recommendations on therapeutic individualization [15]. PPK models can be used to predict drug concentrations, either a priori or a posteriori, via Bayesian estimation procedures, given the measurement of one or more concentrations. However, using previously developed PPK models in the obese patient population requires validation of their predictive ability, as they were not initially developed in this population.

Therefore, we sought to determine whether current knowledge about the PK of apixaban and rivaroxaban, as formalized in published PPK models developed in obese and non-obese patient populations treated for various indications, was applicable to obese patients treated for VTE. To this end, we performed a systematic review of published apixaban and rivaroxaban PPK models and a subsequent external evaluation on an independent dataset to validate their predictive performance in the population of obese patients treated for VTE.

## 2. Materials and Methods

### 2.1. Review of Published PPK Models

The Scopus, MEDLINE, and Embase databases were searched to identify published PPK analyses of rivaroxaban or apixaban using the following keywords: “rivaroxaban” [AND] “pharmacokinetics” for rivaroxaban, and “apixaban” [AND] “pharmacokinetics” for apixaban. The Cochrane database was searched less stringently, using only the keywords “rivaroxaban” or “apixaban”. The reference lists of identified articles were manually screened for additional relevant studies. The search included studies published in English or French between the inception of the databases and 1 November 2020. The included PPK models were those developed using the following: (1) human DOAC data from adult patients; (2) a compartmental, parametric modeling approach. The PPK models were excluded if (1) the model description was insufficient/inadequate to fully reproduce the model, or (2) the model was a physiological-based pharmacokinetic (PBPK) model.

### 2.2. Independent External Validation Data Set

DOAC PK and demographic data were collected from adult patients with obesity (BMI ≥ 30 kg/m²) receiving apixaban or rivaroxaban for VTE treatment and enrolled in a prospective multicenter observational study [16], conducted in the outpatients’ thrombosis clinics of Rennes and Brest University Hospitals between August 2017 and January 2019. Any patient with obesity with VTE followed by the thrombosis center could be included in the study, whatever the time since the initiation of anticoagulation. Apixaban or rivaroxaban plasma concentrations were measured just after the inclusion visit, whatever the time since the last intake, as part of routine care. The exact time between the DOAC intake and blood sampling was strictly recorded, as well as the DOAC dose. No patient in this study was analyzed in previously published DOAC PPK studies. The patients’ age, weight, BMI, gender, and creatinine clearance (calculated using the Cockcroft and Gault formula) were recorded. The plasma apixaban and rivaroxaban concentrations were measured using the commercial assay STA-Liquid-anti-Xa, with specific controls and calibrators on a STA-R Evolution analyzer from STAGO Diagnostica (Asnières sur Seine, France) in each center. The lower limit of quantification (LLOQ) was 20 ng/mL. The study was approved by the local ethics committee. All the patients were informed and did not object to the inclusion.

### 2.3. External Predictive Performance Evaluation of Apixaban and Rivaroxaban PPK Models

The MonolixSuite 2020R1 (Lixoft SAS, Antony, France) was used for external evaluation. The R language and environment (version 4.1.3, R Foundation for Statistical Computing) was used to postprocess the Monolix output and generate graphics. The published PPK models were employed using reported model equations, parameter values, covariate relationships, interpatient variability, parameter covariance, intra-patient variability, and unexplained residual variability. For each model, the apixaban and rivaroxaban concentrations, respectively, were simulated using dosing regimens, sampling times, and covariate information from the external validation dataset (EVD). To assess steady-state concentrations, ten doses were simulated before the first observation. Plasma concentrations below the LLOQ were left censored, according to the M3 method described by Beal [17]. For the models with study-dependent parameter values, an external evaluation was performed for each set of values.

### 2.4. Prediction-Based Diagnostics

Based on the observed concentration (Cobs) and population prediction (Cpred), the prediction error percentage (PE%) and absolute prediction error percentage (APE%) were calculated using the following equations:$$\mathrm{PE}\;\left(\%\right)=\frac{C_{\mathrm{pred}}-{\;C}_{\mathrm{obs}}}{C_{\mathrm{obs}}}\times 100\%\;\mathrm{APE}\;\left(\%\right)=\left|\frac{C_{\mathrm{pred}}-{\;C}_{\mathrm{obs}}}{C_{\mathrm{obs}}}\right|\times 100\%$$
$$\mathrm{APE}\;\left(\%\right)=\left|\frac{C_{\mathrm{pred}}-{\;C}_{\mathrm{obs}}}{C_{\mathrm{obs}}}\right|\times 100\%$$

The median prediction error (MDPE) and the median absolute prediction error (MDAE) were used to evaluate the accuracy and precision of the predictive performance, respectively. The PE% within ±20% (F20) and the PE% within ±30% (F30) were calculated as joint predictors of accuracy and precision. The predictive performance of the candidate models was considered satisfactory if |MDPE| ≤ 20%, MDAE ≤ 30%, F_20 ≥ 35%, and F_30 ≥ 50% [18,19].

### 2.5. Simulation-Based Diagnostics

The predictive performance of each PPK model was evaluated by performing Monte Carlo simulations in Monolix (*n* = 5000) using patient characteristics, dosing, and the sampling scheme from the EVD. The prediction-corrected visual predictive checks (pcVPC) were computed and plotted using the vpc R package, to visually assess if the prediction-corrected simulations generated by a candidate model deviated from the prediction-corrected observed data. The VPC diagnoses both the fixed and random effects in mixed-effects models, helping to determine if the intrapatient and interpatient variability were adequately specified in each model to reproduce the central trend and variability in the EVD. Prediction correction accounts for the differences in dosing and influent patient covariates [20].

The normalized prediction distribution errors (NPDE) were computed with Monolix. Under the null hypothesis that the model under scrutiny describes the EVD, the NPDE should follow the standard normal distribution [21]. Histograms and quantile–quantile plots (QQ plots) of the NPDE were visually inspected for each model.

## 3. Results

### 3.1. Review of Published PPK Studies

The details of the literature search are provided in Figure 1. The dosing information, patient characteristics, and the intended application of the PPK model for each study are described in Table 1.

#### 3.1.1. Apixaban

A total of five PPK models of apixaban were included for external evaluation after the literature retrieval [22,23,24,25,26]. Among them, four were multicenter studies (A1–A4), and one was a single-center study (A5). The sample size of subjects administered apixaban was >100 in all the multicenter studies. Liquid chromatography coupled with mass spectrometry (LC–MS/MS) methods were used to measure the apixaban plasma concentrations in all but one study (A3), which used an anti-Xa chromogenic assay (Table 1). Both intensive and sparse samples were collected in the three studies sponsored by pharmaceutical companies (A1, A2, A4), and only sparse samples were obtained in the other two studies (A3, A5). Table 2 details the models. 

#### 3.1.2. Rivaroxaban

A total of sixteen PPK models of rivaroxaban were included for external evaluation after the literature retrieval [8,24,27,28,29,30,31,32,33,34,35,36,37,38,39,40]. Among them, eleven were multicenter studies (R2–R4, R6–R8, R10, R12–R14, R16), and five were single-center studies (R1, R5, R9, R11, R15). The sample size of the subjects administered rivaroxaban was >100 in all the multicenter studies, and in one single-center study (R1). LC–MS/MS methods were used to measure the rivaroxaban plasma concentrations in twelve studies, and an anti-Xa chromogenic assay was used in four studies (R1, R3, R10, R15). Most of the studies used sparse sampling to collect the PK samples (R1–R4, R8, R10–R16); intensive sampling was used in two studies on healthy subjects (R5, R9); and one study used both sparse and intensive sampling for a subset of patients (R6). Table 2 details the rivaroxaban PPK models. With the exception of one model (R5), all the structural models were one-compartment models. The relative bioavailability was modelled as dose-independent with intersubject variability in four models (R3, R4, R9, R12), as dose-dependent with discrete values according to rivaroxaban intake in four models (R6, R7, R8, R14), and a non-linear dose-dependent relative bioavailability was included in one model (R13). Bodyweight was included as a covariate influencing rivaroxaban PK in ten studies (R1–R4, R8, R10, R11, R13, R14, R16). Total bodyweight was incorporated as a predictor of both rivaroxaban clearance and of the volume of the central compartment in one study (R13). More often, lean body mass (LBM) was used to include the subjects’ weight in the models (R2, R8, R10, R12, R14, R16), although the formulas differed between the studies. Body mass was included indirectly via the estimation of creatinine clearance, using various adaptations of the Cockcroft and Gault formula, in six studies (R1, R3, R4, R10, R13). The other covariates identified were rivaroxaban dose, gender, age, serum creatinine, comedications, HCT, pharmacogenomics, serum alanine aminotransferase (ALT), and blood urea nitrogen (BUN).

### 3.2. External Validation Dataset Cohort

The EVD included 116 PK samples from 69 patients with obesity taking apixaban for VTE treatment, and 121 PK samples from 81 patients with obesity taking rivaroxaban [16]. The median and range of the age and weight of each subpopulation are presented in Table 1. All the patients were sampled after at least one month of treatment. Since the genotypes, HCT, comedications, ALT, and BUN were not collected in our dataset, the values of the parameters affected were arbitrarily set either to median population values to cancel the effect of the covariate, or to credible values for the population (that is ALT = 22 UI/L, HCT = 0.4 for females, HCT = 0.45 for males).

### 3.3. External Predictability Evaluation

#### 3.3.1. Prediction-Based Diagnostics

The accuracy and precision measures generated for each model are provided in Table 3.

##### Apixaban

All two-compartment models and one of the mono-compartment models met the precision and accuracy objectives (A1, A2, A4, A5), regardless of the subpopulation selected for the study-dependent parameter values for A2 and A4 (Figure 2).

The remaining model did not meet any of the objectives, showing insufficient precision and accuracy of this model when applied to the EVD (Table 3).

##### Rivaroxaban

The results indicated an unsatisfactory predictive performance in the prediction-based diagnostics (Table 3). None of the investigated models met all the aforementioned criteria (|MDPE| ≤ 20%, MDAE ≤ 30%, F_20 ≥ 35%, and F_30 ≥ 50%). The values of MDPE, an indicator of predictive accuracy, were within ±20% in ten studies (R2–R4, R6, R7, R10, R13 (with SUB = NVAF), R14–R16) (Figure 3).

The MDAE, an indicator of predictive precision, was less than 30% in two studies (R2, R13 with SUB = NVAF). As a combined predictor of both accuracy and precision, F_20 was over 35% in only one study (R3), and F_30was over 50% in two studies (R2, R13 with SUB = NVAF). Taking both accuracy and precision into account, the studies by Girgis et al. (R2) [28], and Willmann et al. (R13) [8] with SUB = NVAF showed preferable predictive performances compared to the others when applied to the EVD, in which three out of four criteria were met with |MDPE| ≤ 20%, MDAE ≤ 30%, and F_30 ≥ 50%. When selecting SUB = VTE, the model by Willmann et al. (R13) [8] did not perform well.

#### 3.3.2. Simulation-Based Diagnostics

##### Apixaban

The pcVPC of the prediction-corrected plasma apixaban concentrations versus time since the last apixaban dose showed a substantial discrepancy between the observations and simulations (Figure 4), except in two studies (A2, A4).

In particular, two models showed no obvious misspecification (A2 with SUB = ACS, A4 with SUB = non-patient). The NPDE histograms and QQ plots are shown in Figure 5.

##### Rivaroxaban

The pcVPC showed a large discrepancy between the prediction-corrected observations and the simulations in most of the models (Figure 6).

A noticeable trend of over- or under-prediction was observed, except in four models (R2, R13 with SUB = NVAF, R16) which showed slight misspecifications in the early concentrations post-dose. The NPDE histograms and QQ plots confirmed the poor characterization of the rivaroxaban PK of the EVD (Figure 7).

## 4. Discussion

This is the first study to systematically evaluate the predictive performance of published PPK models for apixaban and rivaroxaban by external validation in a population of obese patients treated for VTE. This analysis highlights the value of some models to explore hypothetical scenarios to determine apixaban and rivaroxaban dosing in patients with high body weight or BMI. Based on our analysis, two of the published apixaban PPK models evaluated adequately described the observed anticoagulant PK in the EVD, whereas no published rivaroxaban PPK model did. The PPK models based on studies with relatively large patient populations and an intensive sampling strategy better characterized the DOAC PK of the EVD overall. The sampling including the distribution phase—and not just sampling around the peak and trough—improved the assessment of the number of compartments needed to effectively characterize the DOAC PK.

The two-compartment models best described the pharmacokinetics of apixaban in the external validation dataset. Models A2 and A4 were based on studies that used both intensive and sparse sampling and described the PK of apixaban in the EVD satisfactorily. The mass descriptor included in models A2 and A4 was Total Body Weight (TBW), similar to the all-apixaban PPK models analyzed, and the influence of renal function on apixaban PK was included using the Cockcroft formula (model A4), or the Cockcroft formula limited to 150 mL/min (model A2). Because these approaches were common to all the apixaban models analyzed, it was not possible to conclude whether other descriptors of weight (e.g., Lean Body Weight (LBW), adjusted weight) or renal function (e.g., serum creatinine, CKD-epi) would be of any value. The choice of method for estimating renal function has been cited as a cause of dosing discrepancies; however, the ELIQUIS package insert refers to renal function only in the context of renal failure; hence, no impact is expected with a cutoff of 150 mL/min on the Cockcroft estimate.

Although similarly bi-compartmental, and developed from rich data, the A1 model did not satisfactorily describe the EVD, as revealed by the pcVPC and the NPDE residuals inspection. The A1 model differed mainly from the other two-compartment apixaban models (A2, A4) by the absence of a dose-dependent bioavailability model. Not unexpectedly, although the interindividual variability associated with the absorption constant, Ka, was comparable between these three models, the discrepancies between the A1 model simulations and observed concentrations objectified by the pcVPCs were mainly observed soon after the apixaban dosing.

Notwithstanding their large number, no published rivaroxaban PPK models characterized the EVD adequately. Two models performed better than the others, without meeting all the validity criteria (model R2, R13). Both were one-compartment models, derived from sparse sampling studies. Otherwise, the models were different. Model R13 included a dose-dependent bioavailability model with interindividual variability in the absorption constant, Ka. The influence of weight was integrated with TBW, and the influence of renal function on rivaroxaban clearance was modeled by a modified version of Cockcroft clearance (referred to as “Tietz-truncated clearance” by the authors). The R2 model described absorption with an absorption constant, Ka, without interindividual variability. The effect of body mass on rivaroxaban PK was included via LBW, and the effect of renal function on anticoagulant clearance was included using serum creatinine. The different approaches of the two best-performing rivaroxaban PPK models did not allow for a conclusion as to which strategy was best overall. There was no trend toward better model performance with respect to the number of subjects included in the studies, the number of samples collected, the study population (e.g., healthy volunteers, orthopedic surgery patients, patients with NVAF), or the magnitude of the residual error (although nine models had a proportional error greater than 40%).

Of the papers reviewed, two studies were notable for their common objective: to identify the influence of body mass on rivaroxaban PK (R1, R10). The study by Speed et al. (R10) [35] was a large study with a sparse sampling strategy. The influence of body mass on PK was included with LBM, and the impact of renal function on rivaroxaban clearance was incorporated by Cockcroft creatinine clearance calculated with LBM instead of TBW. In the model developed by Barsam et al. (R1) [27], the only covariate retained was the influence of Cockcroft creatinine clearance, calculated with TBW, on rivaroxaban clearance. However, the performance of neither model was satisfactory when applied to EVD. The underestimation (R1) or overestimation (R10) of the simulations compared with the observed concentrations, along with the high bias and imprecision identified through our analysis, did not allow us to confirm the limited impact of weight on rivaroxaban PK suggested by the authors.

The appropriateness of the external data set to assess each model needs consideration in light of the high degree of bias and imprecision found across most of the published models. The dosing in the observational prospective study [16] used to generate the EVD was standard, comparable with the dosing described in the published PPK models and is an unlikely source of bias. An additional limitation was the lack of information in the dataset regarding some of the covariates in the published models. The introduction of bias due to the imputation of missing variables (HCT for models A4 and R4, BUN for model R12, pharmacogenomics for models A5 and R15) cannot be excluded. Among these covariates, race is a special case (models A1, A2). Firstly, the collection of ethnic statistics is not allowed in France, and, secondly, the concept of race is questionable and an inadequate descriptor of the distribution of genetic variability in our species [41]. In the specific context of PK, a considerable amount of literature comparing PK between ethnic groups has been produced, with more papers reporting similarities than differences, and a decrease in the proportion of papers showing ethnic differences over time [42]. Overall, these data suggest disregarding racial covariates. The EVD anticoagulant assay could also be questioned, but the correlation between drug-calibrated anti-FXa methods and LC–MS/MS has been demonstrated for routine concentrations [43], and the satisfactory performance of some apixaban PPK models developed from LC–MS/MS data (A2, A4) is reassuring. Finally, as all the models were implemented in Monolix software, the implementation of the residual error models might have introduced additional imprecision in the model evaluations because of its difference with the implementation of NONMEM [44].

In light of the reassuring clinical data, the latest ISTH guidance suggested using standard doses of apixaban or rivaroxaban in the treatment of VTE, regardless of patient weight or high BMI [14]. Nevertheless, appropriate PPK models for this population of patients could be invaluable in bridging the gap between analytical chemistry and patient clinical data.

For apixaban, the results of systematic external evaluations supported a satisfactory description of its PK in the obese patient population treated for VTE by two published PPK models, using TBW as a descriptor of body mass (A2, A4). In contrast, for rivaroxaban, the lack of alignment of the published PPK analyses with the aim of adequately describing rivaroxaban PK in the obese patient population, combined with their poor predictive performance in this population, emphasized the need to develop such a PPK model. The development of this PPK model should be based on a well-powered study in the population of interest, with an intensive PK sampling strategy to identify clinically relevant covariates that can adequately characterize rivaroxaban PK. Since our last literature search, new pharmacokinetic studies have been conducted or are underway. For example, a Pubmed/MEDLINE search in February 2023 using the keywords “rivaroxaban population pharmacokinetics” for 2021–2023 identified five studies [45,46,47,48,49] that would have met our systematic review inclusion criteria by title and abstract. All were single-compartment models with first-order uptake, and none were specifically designed for the obese patient population. The present study did not evaluate these models from studies conducted after our last literature search, but an ongoing phase 1 study aiming to assess the PK of rivaroxaban used as a therapeutic anticoagulant dose in patients with previous bariatric surgery, and in morbidly obese subjects (NCT04180436) [50], will provide the data necessary to develop an adequate PPK model in this population.

## 5. Conclusions

The study was not designed to draw clinical conclusions about the management of obese patients with VTE. However, the analysis provided information on the pharmacokinetics of the direct oral anticoagulants studied. Several population pharmacokinetic models of apixaban were applicable to the population of obese patients receiving curative apixaban treatment for VTE, suggesting that the results of these models developed in the general population are relevant to this specific population. In contrast, none of the models evaluated for rivaroxaban were applicable to obese patients treated for VTE. Extrapolations from these models, their parameter values, or their simulation results should not be applied to obese patients treated with rivaroxaban for VTE.

## Figures and Tables

**Figure 1 pharmaceutics-15-00665-f001:**
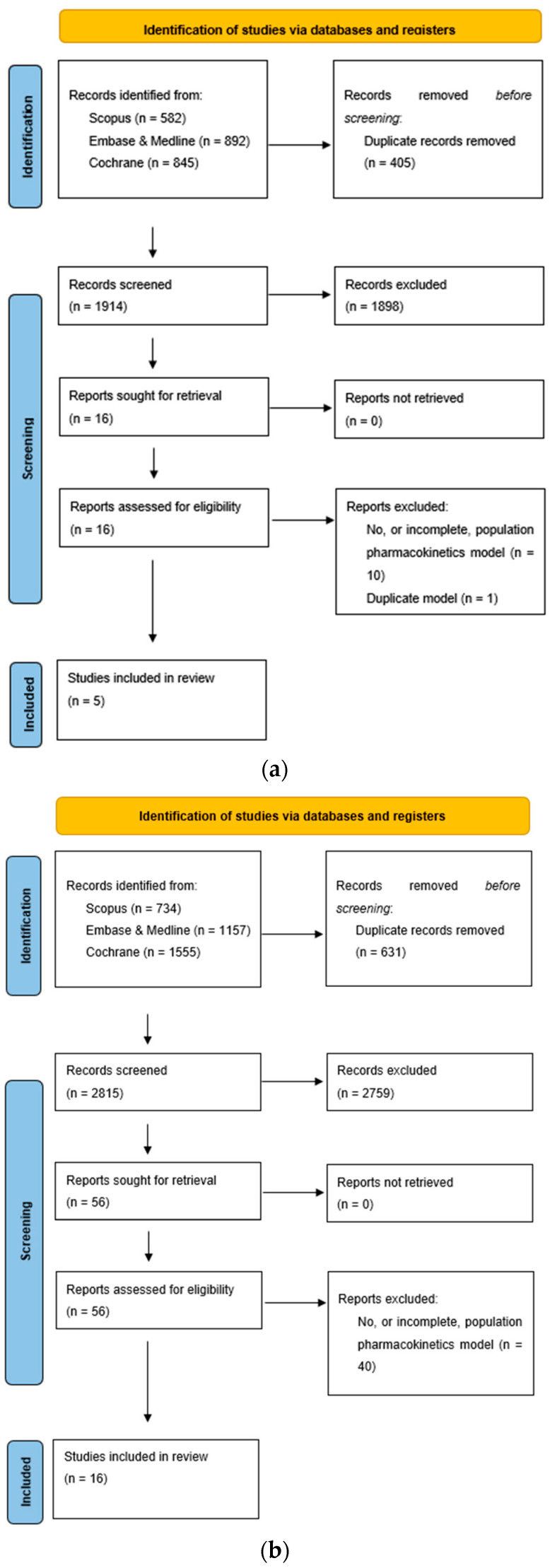
(**a**) PRISMA flowchart for apixaban population pharmacokinetic studies; (**b**) PRISMA flowchart for rivaroxaban pharmacokinetic studies.

**Figure 2 pharmaceutics-15-00665-f002:**
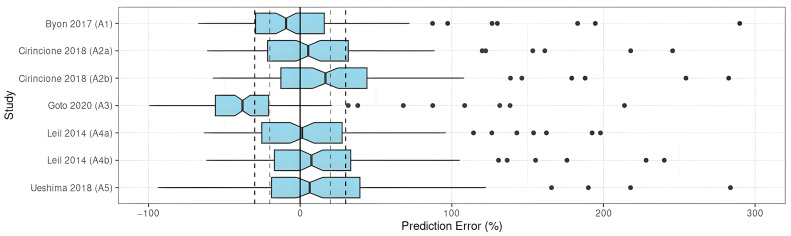
PE% of apixaban population pharmacokinetic studies applied to the external validation dataset [22,23,24,25,26].

**Figure 3 pharmaceutics-15-00665-f003:**
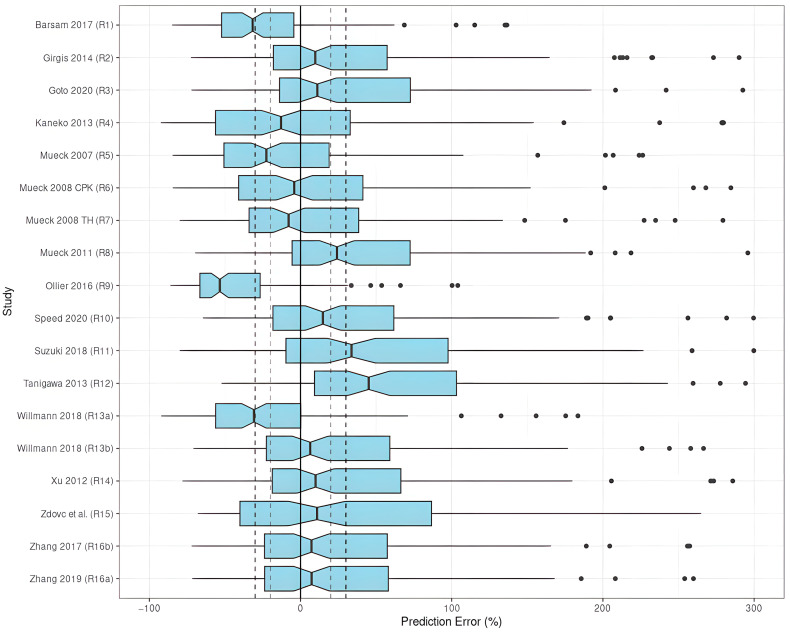
PE% of rivaroxaban population pharmacokinetic studies applied to the external validation dataset [8,24,27,28,29,30,31,32,33,34,35,36,37,38,39,40].

**Figure 4 pharmaceutics-15-00665-f004:**
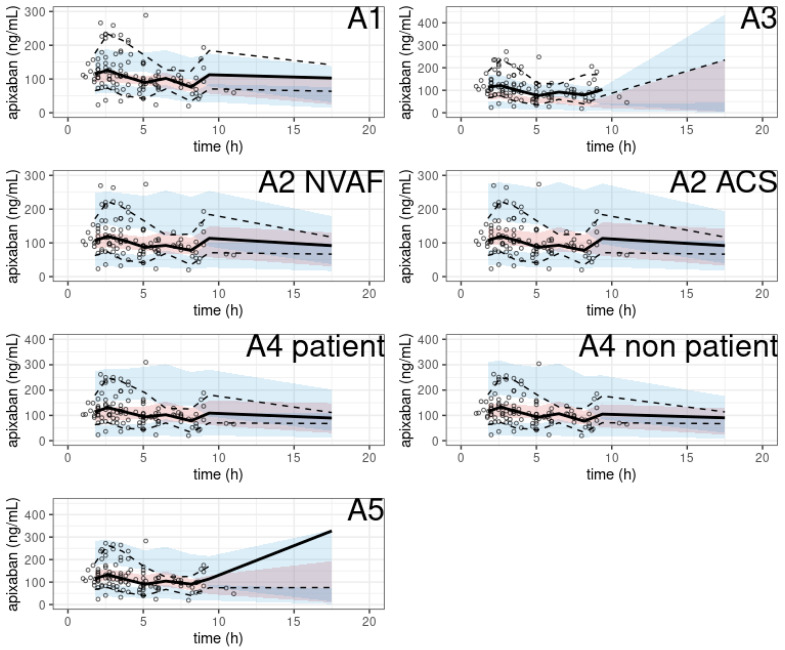
pcVPC of apixaban population pharmacokinetic studies applied to the external validation dataset.

**Figure 5 pharmaceutics-15-00665-f005:**
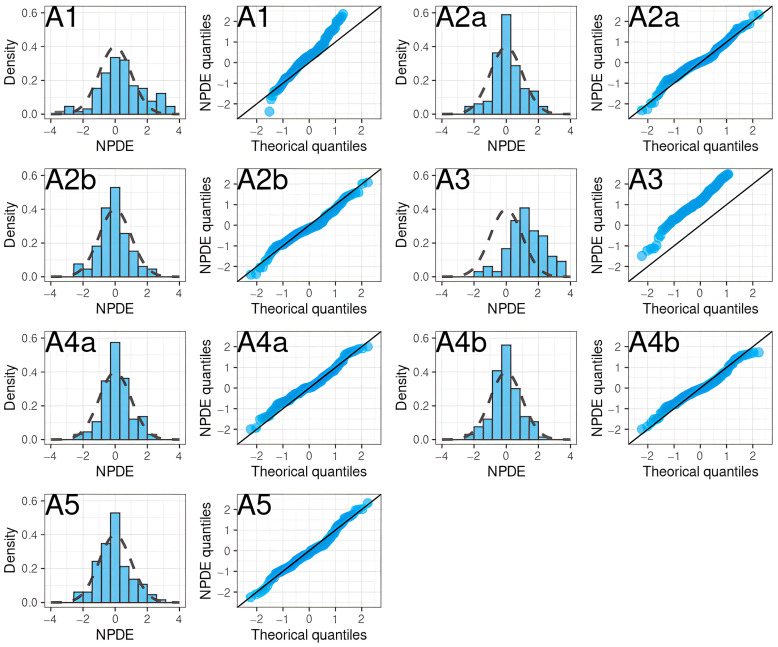
NPDE histograms and QQ plots of apixaban population pharmacokinetic studies applied to the external validation dataset.

**Figure 6 pharmaceutics-15-00665-f006:**
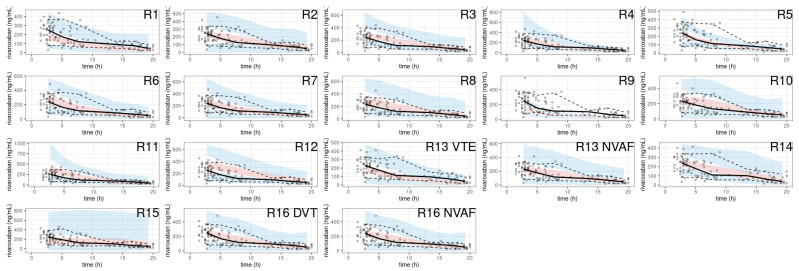
pcVPC of rivaroxaban population pharmacokinetic studies applied to the external validation dataset.

**Figure 7 pharmaceutics-15-00665-f007:**
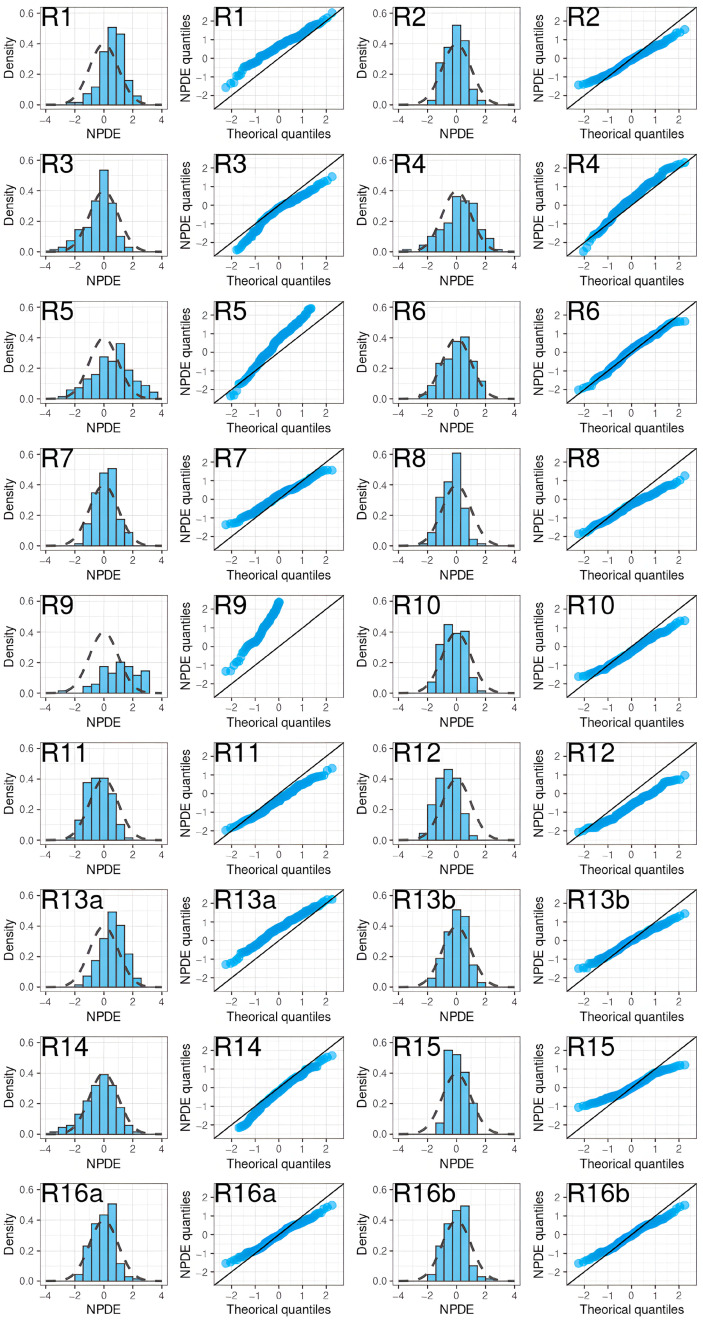
NPDE histograms and QQ plots of rivaroxaban population pharmacokinetic studies applied to the external validation dataset.

**Table 1 pharmaceutics-15-00665-t001:** Design of the selected population pharmacokinetic studies.

Model Reference	PK Study Reference	N Patients	Age	Weight	Daily Dose (mg)	Dosing Frequency	N PK Samples	Sampling Regimen	Assay	Intended Application of the PPK Model
EVD apixaban	Ballerie 2021 [16]	69	55 (20–86)	99 (79–150)	2.5, 5	BID	116	Sparse	Anti-Xa chromogenic assay LLOQ 20 ng/mL	No PPK model
A1	Byon 2017 [22]	970	(18–89)	167 patients > 100 kg	2.5–50	single dose, QD, BID	8323	Intensive + sparse	LC-MS/MS LLOQ 1 ng/mL	PKPD EER analysis in patients with VTE
A2	Cirincione 2018 [23]	4385	68 (18–94)	81.4 (32–198.2)	2.5–50	single dose, QD, BID	11,968	Intensive + sparse	LC-MS/MS LLOQ 1 ng/mL	Explain PK heterogeneity in patients with NVAF
A3	Goto 2020 [24]	140	79.1 * ± 5.8 (2.5 mg BID) 70.9 * ± 7.5 (5 mg BID)	55.7 * ± 10.6 (2.5 mg BID) 62.8 * ± 11.7 (5 mg BID)	2.5, 5	BID	183	Sparse	Anti-Xa chromogenic assay LLOQ 20 ng/mL	Compare anti-Xa DOAC PK
A4	Leil 2014 [25]	1284	NA	NA	2.5–50	single dose, QD, BID	11,252	Intensive + sparse	LC-MS/MS LLOQ 1 ng/mL	PKPD EER analysis in patients undergoing orthopedic surgery
A5	Ueshima 2018 [26]	81	68 (40–85)	65 (41–92)	5–20	BID	276	Sparse	LC-MS/MS LLOQ 2.5 ng/mL	Explain PK heterogeneity in patients with NVAF
EVD rivaroxaban	Ballerie 2021 [16]	81	64.5 (20–85)	102 (73.0–178)	20	QD	121	Sparse	Anti-Xa chromogenic assay LLOQ 20 ng/mL	No PPK model
R1	Barsam 2017 [27]	101	52 * (20–86)	88 * ± 23.4	10–30	QD, BID	193	Sparse	Anti-Xa chromogenic assay LLOQ 20 ng/mL	Study the impact of weight on rivaroxaban PK
R2	Girgis 2014 [28]	161	NA	NA	15–20	QD	801	Sparse	LC-MS/MS LLOQ 0.5 ng/mL	Confirm dose selection in patients with NVAF
R3	Goto 2020 [24]	119	73.1 * ± 10.0 (10 mg QD) 66.7 * ± 10.0 (15 mg QD)	60.3 * ± 15.5 (10 mg QD) 67.3 * ± 13.8 (15 mg QD)	10, 15	QD	162	Sparse	Anti-Xa chromogenic assay LLOQ 20 ng/mL	Compare anti-Xa DOAC PK
R4	Kaneko 2013 [29]	597	72 (34–89)	63.9 (35–104)	10, 15	QD	1834	Sparse	LC-MS/MS LLOQ 0.5 ng/mL	Confirm dose selection in Japanese patients with NVAF
R5	Mueck 2007 [30]	43	33 * (20–45)	NA	5–60	QD, BID	1809	Intensive	LC-MS/MS LLOQ 0.5 ng/mL	Describe rivaroxaban PK in healthy subjects
R6	Mueck 2008 CPK [31]	1009	65 (26–87) (hip study) 67 (39–92) (knee study)	76 (45–125) (hip study) 86 (50–173) (knee study)	5–60	QD, BID	7568	Intensive + sparse	LC-MS/MS LLOQ 2.5 ng/mL	Describe rivaroxaban PK in patients undergoing major orthopaedic surgery
R7	Mueck 2008 TH [32]	758	66 (26–93)	75 (45–120)	5–20	QD, BID	5743	Sparse	LC-MS/MS LLOQ 2.5 ng/mL	Compare the PKPD of QD and BID rivaroxaban in patients undergoing total hip replacement
R8	Mueck 2011 [33]	870	61 (18–94)	85 * ± 17 (male) 73 * ± 16 (female)	10–60	QD, BID	4634	Sparse	LC-MS/MS LLOQ 2.5 ng/mL	Describe rivaroxaban PK in patients treated for acute DVT and simulate exposure in patients with NVAF
R9	Ollier 2016 [34]	12	26 (20–30)	71 (62–88)	40	Single dose	192	Intensive	LC-MS/MS LLOQ 5 ng/mL	Study the effect of activated charcoal on rivaroxaban absorption
R10	Speed 2020 [35]	913	67.0 * ± 15.0	85.8 * ± 23.1	15–30	QD, BID	1108	Sparse	Anti-Xa chromogenic assay LLOQ 20 ng/mL	Understand the influence of WT on rivaroxaban PK
R11	Suzuki 2018 [36]	96	68.0 * ± 9.5	69.1 ± 11.4	10–15	QD	192	Sparse	LC-MS/MS LLOQ 1 ng/mL	Describe rivaroxaban PK in Japanese patients with NVAF
R12	Tanigawa 2013 [37]	182	65.6 (30–92)	67.2 (45–103)	5–40	QD, BID	842	Sparse	LC-MS/MS LLOQ 0.5 ng/mL	Select dose for Japanese patients with NVAF
R13	Willman 2018 [8]	4918	60.5 * ± 11.8	82.5 * ± 16.9	5–60	QD, BID	22,843	Sparse	LC-MS/MS LLOQ 0.5 ng/mL	Describe rivaroxaban PK across multiple patient populations
R14	Xu 2012 [38]	2290	57 (24–87)	84 (36–181)	5–20	QD, BID	6644 **	Sparse	LC-MS/MS LLOQ 0.5 ng/mL	Describe rivaroxaban PKPD in patients with ACS
R15	Zdovc 2019 [39]	17	64 (49–82)	84 (54–125)	10	QD	82	Sparse	Anti-Xa chromogenic assay LLOQ 1 ng/mL	Investigate the influence of ABCB1 polymorphism on rivaroxaban PKPD
R16	Zhang 2017 [40]	285	59 (31–83) (DVT study) 65 (51–81) (NVAF study)	54.1 (40.1–72.7) (DVT study) 56.6 (42.5–73.6) (NVAF study)	20–40	QD	NA	Sparse	LC-MS/MS LLOQ 0.5 ng/mL	Evaluate the effect of food on rivaroxaban PK

EVD: external validation data set, BID: twice a day, QD: once a day, LLOQ: lower limit of quantification, ERR: exposure–response relationship, NVAF: non-valvular atrial fibrillation, LC–MS/MS: liquid chromatography with tandem mass spectrometry, DVT: deep vein thrombosis, WT: weight, ACS: acute coronary syndrome, ABCB1: ATP-binding cassette subfamily member 1, PK: pharmacokinetics, PKPD: pharmacokinetics-pharmacodynamics, NA: not available. * mean value, ** number of paired PKPD observations.

**Table 2 pharmaceutics-15-00665-t002:** Details of the published population pharmacokinetic models.

Model Reference	Modeling Software	Structural Model	Relative Bioavailability	Parameter Values	Covariates	Interpatient Variability	Residual Error
A1	NONMEM 7.2	2 CMT	NA	Ka (1/h) = 0.440 CL (L/h) = 4.35 Vc (L) = 32.1 Q (L/h) = 1.62 Vp (L) = 19.8	Ka: evening dosing CL: Sex, WT, CrCL *, Race, INH Vc: WT	ωKa = 0.474 ωCL = 0.322 ωVc = 0.232	Additive
A2	NONMEM 7.1	2 CMT	I50 = −0.322 Gamma = 0.857	Ka (1/h) = 0.473 CL (L/h) = 3.59 Vc (L) = 30.0 Q (L/h) = 1.91 Vp (L) = 27.0	Ka: AMPM CL: CrCL *, Age, Sex, Race, INH, SUB Vc: WT, SUB	ωKa = 0.513 ωK = 0.309 ωk12 = 1.245 ωk21 = 0.490 ωVc = 0.172	Proportional = 0.31
A3	Phoenix NLME 8.1	1 CMT	NA	Ka (1/h) = 0.42 CL (L/h) = 4.74 Vc (L) = 30	CL: CrCL	ωCL = 0.266 ωVc = 0.566	Proportional = 0.34
A4	NONMEM 6.1.1	2 CMT	ED50 = 32.5 Imax = 0.705 Gamma = 2.21	Ka (1/h) = 0.188 CL (L/h) = 4.75 ** Vc (L) = 22.9 Q (L/h) = 2.60 Vp (L) = 22.2	Ka: SUB CL: Age, Sex, Dose ***, CrCL, Vc: WT, HCT	ωKa = 0.532 ωCL = 0.375 ωVc = 0.252 ωQ = 0.491 ωVp = 0.735 correlation ωVc ωCL = 0.915	Proportional = 0.34 Additive = 3.38
A5	NONMEM 7.3.0	1 CMT	NA	Ka (1/h) = 0.42 CL (L/h) = 1.53 Vc (L) = 24.7	CL: CrCL, PGx	ωCL = 0.266 ωVc = 0.566	Proportional = 0.34
R1	NONMEM 7.2.13	1 CMT	NA	Ka (1/h) = 1.21 CL (L/h) = 8.86 Vc (L) = 101	CL: CrCL	ωCL = 0.480 ωVc = 0.600	Proportional = 0.31
R2	NONMEM 7.10	1 CMT	NA	Ka (1/h) = 1.16 CL (L/h) = 6.10 Vc (L) = 79.7	CL: Age, SCre Vc: LBM, Age	ωCL = 0.342 ωVc = 0.175	Proportional = 0.479
R3	Phoenix NLME 8.1	1 CMT	F1 = 1	Ka (1/h) = 0.617 CL (L/h) = 5.59 Vc (L) = 50.9	CL: CrCL	ωKa = 0.540 ωCL = 0.394 ωVc = 0.583 ωF1 = 0.365	Proportional = 0.131
R4	NONMEM 6.2.0	1 CMT	F1 = 1	Ka (1/h) = 0.617 CL (L/h) = 4.73 Vc (L) = 43.8	CL: CrCL, HCT	ωKa = 0.582 ωCL = 0.410 ωVc = 0.636 ωF1 = 0.377 correlation ωVc ωCL = 0.729	Proportional = 0.131
R5	NONMEM 5.1.1	2 CMT	NA	Tlag (h) = 0.25 Ka (1/h) = 0.97 CL (L/h) = 9.17 Vc (L) = 55.3 Q (L/h) = 1.35 Vp (L) = 12.6	Vc: Dose Vp: Dose	IOV Tlag = 0.847 ωKa = 0.497 IOV Ka = 0.794 ωCL = 0.173 ωVc = 0.300 ωVp = 0.373	Proportional = 0.254
R6	NONMEM 5.1.1	1 CMT	F1 = 1	Ka (1/h) = 1.81 CL (L/h) = 7.3 Vc (L) = 49.1	F1: Dose	ωCL = 0.373	Proportional = 0.371
R7	NONMEM 5.1.1	1 CMT	F1 = 1	Ka (1/h) = 1.49 CL (L/h) = 7.51 Vc (L) = 58.2	F1: Dose	ωCL = 0.369 ωVc = 0.316	Proportional = 0.526
R8	NONMEM 5.1.1	1 CMT	F1 = 1	Ka (1/h) = 1.23 CL (L/h) = 5.67 Vc (L) = 54.4	F1: Dose CL: Age, SCr Vc: LBM, Age	ωCL = 0.384 ωVc = 0.282	Proportional = 0.407
R9	Monolix 4.3.2	1 CMT	F = 0.569	f1 = 0.748 f2 = 0.348 Tmax1 (h) = 0.274 dTmax2 (h) = 1.94 dTmax3 (h) = 11.5 CV1 = 0.495 CV2 = 0.167 CV3 = 0.651 CL (L/h)= 7.4 Vc (L) = 28.4	Activated charcoal effect on input rate	ωF = 0.253 IOV F = 0.728 IOV f1 = 0.997 IOV correlation ωF ωf1 = −0.717 ωCV1 = 0.570 ωVc = 0.085	Proportional = 0.194
R10	NONMEM 7.4.2	1 CMT	NA	Ka (1/h) = 0.707 CL (L/h) = 5.57 Vc (L) = 59.4 Lambda = −1.83	CL: CrCl ***** Vc: LBM ******	ωCL = 0.227 *******	Proportional = 0.4637
R11	Phoenix NLME 1.4	1 CMT	NA	Ka (1/h) = 1.37 CL (L/h) = 4.40 Vc (L) = 38.2	CL: CrCL, ALT, INH	ωKa = 0.426 ωCL = 0.204 ωVc = 0.583	Proportional = 0.418
R12	NONMEM 5.1.1	1 CMT	F1 = 1	Ka (1/h) = 0.60 CL (L/h) = 4.72 Vc (L) = 42.9	CL: BUN	ωF1 = 0.244 ωKa = 0.680 ωCL = 0.213	Proportional = 0.402
R13	NONMEM 7.3	1 CMT	Fmin = 0.590 Fmax = 1.25 D50 = 14.4	Ka (1/h) = 0.821 CL (L/h) = 6.58 Vc (L) = 62.5	CL: CrCL, WT, INH, SUB Vc: WT, Age, Sex	ωKa = 0.792 ωCL = 0.409 ωVc = 0.198 correlation ωCL ωVc = 0.834	Proportional = 0.451
R14	NONMEM 6.1.1	1 CMT	F1 = 1	Ka (1/h) = 1.24 CL (L/h) = 6.48 Vc (L) = 57.9	F: Dose CL: Age, SCr **** Vc: LBM, Age	ωKa = 1.037 ωCL = 0.306 IOV CL = 0.316 ωVc = 0.010	Additive = 0.352
R15	NONMEM 7.3	1 CMT	NA	Ka (1/h) = 0.147 CL (L/h) = 6.12 Vc (L) = 96.8	CL: PGx	ωKa = 2.004 ωCL = 0.709	Proportional = 0.595
R16	NONMEM 7.2	1 CMT	F1 = 1	Ka (1/h) = 0.982 CL (L/h) = 6.31 Vc (L) = 70.3	F1: SUB CL: Age, SCr **** Vc: LBM, Age	ωCL = 0.336 ωVc = 0.154	Proportional = 0.475

CMT: compartment, NA: not available, Ka: absorption rate constant, CL: clearance, Vc: volume of the central compartment, Q: intercompartmental clearance, Vp: volume of the peripheral compartment, WT: weight, CrCL: creatinine clearance, ω: standard deviation of the random effect, I50: logit for reduction in F at 50 mg, Gamma: shape parameter for F, AMPM: morning or evening dosing, INH: enzymatic inhibitor, SUB: subpopulation of the study, ED50: dose at which half of the maximal reduction in F is achieved, Imax: maximum reduction in relative bioavailability, PGx: pharmacogenomics, Scre: serum creatinine, LBM: lean body mass, F1:bioavailability, HCT: hematocrit, Tlag: lag time for drug absorption, IOV: inter-occasion variability, F: bioavailability, f1, f2, Tmax1: time the first inverse Gaussian function reaches its maximum, dTmax2: time delta from Tmax1 for the second inverse Gaussian function to reach its maximum, dTmax3: time delta from Tmax2 for the third inverse Gaussian function to reach its maximum, CV1: coefficient of variation of the first inverse Gaussian function, CV2: coefficient of variation of the second inverse Gaussian function, CV3: coefficient of variation of the third inverse Gaussian function, Lambda: Box–Cox transformation parameter, ALT: alanine aminotransferase, BUN: blood urea nitrogen, Fmin, Fmax and D50: parameters for relative bioavailability as a function of dose. * capped at 150 mL/min, ** CLNR + CLRmax in Emax model for CL, *** Dose > 25 mg, **** Serum creatinine in mg/dL, ***** using LBM, ****** LBM formula different from Mueck formula, ******* Box–Cox transformed. The structural models were two-compartment models for studies with both intensive and sparse sampling, and one compartment models for studies with sparse sampling (A3, A5). Non-linear dose-dependent relative bioavailability was modelled using a power model in two studies (A2, A4). Bodyweight was included as a covariate influencing apixaban PK in all the studies. Total body weight was incorporated directly as a predictor of the volume of the central compartment in three studies, and indirectly through an estimation of creatinine clearance by Cockcroft and Gault’s formula, as a predictor of apixaban clearance in all the studies. Other covariates identified were the time (afternoon, evening, morning) and dose of the last apixaban intake, gender, age, race, comedications (CYP450 inhibitors), hematocrit (HCT), and pharmacogenomics.

**Table 3 pharmaceutics-15-00665-t003:** Bias and imprecision of the published of the published population pharmacokinetic models applied to the EVD.

	MDPE (%)	MDAE (%)	F_20_ (%)	F_30_ (%)
A1 (Byon 2017 [22])	−7.8	25.0	42.2	56.0
A2 (Cirincione 2018 SUB = ACS [23])	17.0	29.0	37.1	51.7
A2 (Cirincione 2018 SUB = NVAF [23])	5.5	24.8	41.4	55.2
A3 (Goto 2020 [24])	−38.0	39.7	16.4	31.9
A4 (Leil 2014 SUB = patients [25])	1.7	27.3	37.1	52.6
A4 (Leil 2014 SUB = non patients [25])	7.6	27.2	39.7	56.0
A5 (Ueshima 2018 [26])	7.2	25.8	40.5	54.3
R1 (Barsam 2017 [27])	−31.5	39.6	21.5	34.7
R2 (Girgis 2014 [28])	9.7	29.9	33.9	50.4
R3 (Goto 2020 [24])	11.6	34.0	36.4	45.4
R4 (Kaneko 2013 [29])	−12.2	45.8	26.4	33.9
R5 (Mueck 2007 [30])	−22.2	41.2	24.0	39.7
R6 (Mueck 2008 CPK [31])	−3.4	41.7	25.6	38.8
R7 (Mueck 2008 TH [32])	−7.3	36.0	25.6	42.1
R8 (Mueck 2011 [33])	24.8	35.4	34.7	43.8
R9 (Ollier 2016 [34])	−53.4	53.9	14.9.	20.7
R10 (Speed 2020 [35])	17.3	32.2	32.2	47.9
R11 (Suzuki 2018 [36])	36.2	46.4	19.8	33.1
R12 (Tanigawa 2013 [37])	49.0	49.4	25.6	35.5
R13 (Willman 2018 SUB = VTE [8])	−30.4	42.7	23.1	33.1
R13 (Willman 2018 SUB = NVAF [8])	6.6	28.5	32.2	51.2
R14 (Xu 2012 [38])	10.9	37.5	30.6	46.2
R15 (Zdovc 2019 [39])	18.6	51.8	19.0	24.8
R16 (Zhang 2017 SUB = DVT [40])	7.6	30.1	25.6	48.8
R16 (Zhang 2017 SUB = NVAF [40])	8.5	31.0	25.6	48.8

SUB: subpopulation of the study, ACS: acute coronary syndrome, NVAF: non-valvular atrial fibrillation, VTE: venous thromboembolism, DVT: deep vein thrombosis.

## Data Availability

No new data were created or analyzed in this study. Data sharing is not applicable to this article.

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
