# Peer review of "Transferability of Published Population Pharmacokinetic Models for Apixaban and Rivaroxaban to Subjects with Obesity Treated for Venous Thromboembolism: A Systematic Review and External Evaluations"

_pharmaceutics, 2023, doi:10.3390/pharmaceutics15020665_

Round 1

Reviewer 1 Report

We consider the manuscript very pertinent to the ground of this Journal, containing crucial information about apixaban and rivaroxaban PPK models ‘validation for obese populations. We found it particularly significant to include and discuss the lack of success of rivaroxaban PK models for the obese patients’ group, evidencing that the kinetics of this drug needs to be thoroughly studied in obese patients.  

The objectives of this work are clearly stated and comprehensively justified.

The introduction covers both old and new references concisely and perfectly integrates the theme's main aspects.

This article is well written, with a good organization of the contents. The whole experimental design, to test the transferability of the PPK models and the external evaluations, was carefully elaborated and meticulously presented, integrating the main aspects of the scenario under study. Furthermore, the adjacent statistical modeling concerns were appropriately studied and verified throughout the manuscript, e.g. accuracy and precision of the predictive performance of the candidate models.

The cited core references are recent and appropriate to the discussion. The manuscript is very nicely discussed but seems to lack the Conclusions item. Please make a clear distinction between Discussion and Conclusions.

Regarding the presentation of the results, we found it suitable, clear, and concisely written ensuring a proper interpretation and understanding, though some suggestions will be done as specific comments.

Specific comments:

#1_Table 1_ L244 and Table 2_L 252_ the captions of these tables seem incomplete for both. We suggest the authors include a very concise description of the abbreviations used for model (Tab. 1+Tab. 2), dosing frequency (Tab. 1), and parameters/covariates (Tab. 2). The description of the asterisks used seem to be missing as well, for both Tables.

 If these abbreviations/ captions are presented (e.g. at the end of the Table), it´s easier for the reader to follow the discussion and interpret the results. 

Author Response

We would like to thank the reviewers for taking the time and effort necessary to review the manuscript. We sincerely appreciate all valuable comments and suggestions, which helped us to improve the quality of the manuscript.

Reviewer #1

  • A final paragraph has been added to clarify the message at the end of the paper, in line with the reviewer's recommendations:

“5. Conclusion

The study was not designed to draw clinical conclusions about the management of obese patients with VTE. However, the analysis provided information on the pharmacokinetics of the direct oral anticoagulants studied. Several population pharmacokinetic models of apixaban were applicable to the population of obese patients receiving curative apixaban treatment for VTE, suggesting that the results of these models developed in the general population are relevant to this specific population. In contrast, none of the models evaluated for rivaroxaban were applicable to obese patients treated for VTE. Extrapolations from these models, their parameter values or their simulation results should not be applied to obese patients treated with rivaroxaban for VTE.”

  • The missing abbreviations and captions have been added to the tables 1 to 3, we thank the reviewer for identifying this oversight on our part.

Reviewer 2 Report

Apixaban and rivaroxaban are important drugs, and the data in this paper have reference value for the use of this drug. The writing of the introduction part can be simplified. A conclusion can be briefly stated at the end.

Author Response

We would like to thank the reviewers for taking the time and effort necessary to review the manuscript. We sincerely appreciate all valuable comments and suggestions, which helped us to improve the quality of the manuscript.

Reviewer #2

A final paragraph has been added to clarify the message at the end of the paper, in line with the reviewer's recommendations:

“5. Conclusion

The study was not designed to draw clinical conclusions about the management of obese patients with VTE. However, the analysis provided information on the pharmacokinetics of the direct oral anticoagulants studied. Several population pharmacokinetic models of apixaban were applicable to the population of obese patients receiving curative apixaban treatment for VTE, suggesting that the results of these models developed in the general population are relevant to this specific population. In contrast, none of the models evaluated for rivaroxaban were applicable to obese patients treated for VTE. Extrapolations from these models, their parameter values or their simulation results should not be applied to obese patients treated with rivaroxaban for VTE.”

Reviewer 3 Report

I wish to warmly congratulate the Authors, since their effort was so huge! The manuscript has been well conceived and reported, some comments:

- generally speaking, either you prepare a systematic review or an external validation trial; these topics should be independently treated, IMHO. Could you kindly explain why you have marched against these classical Cochrane concepts?

- last literature query: it should be updated, since the last one comes from Nov 2020, a bit out of date

- line 22, PPK is undefined (population pharmacokinetics, I suppose)

- line 478, no new data analyzed? well, how to define those used for the systematic review external validation? It seems that those observational data have been reanalyzed: could you please better detail this topics?

- discussion: even well written and detailed, there's a general lackness for a clear take-home message. What readers should do in the future, at the light of your ideas and conclusions!? Maybe nothing, if results shoyld be considered as unconclusive!

Author Response

We would like to thank the reviewers for taking the time and effort necessary to review the manuscript. We sincerely appreciate all valuable comments and suggestions, which helped us to improve the quality of the manuscript.

Reviewer #3

  • The reviewer correctly points out that systematic review and external model validation are usually different methodologies and are treated separately. However, the approach we took in this paper is not new to the field [see references 1-3 below]. Our aim was to assess whether current knowledge of the pharmacokinetics of apixaban and rivaroxaban, in the form of population pharmacokinetic models, is appropriate and sufficient to inform pharmacokinetics in our population of interest: obese patients treated for VTE. The first step was to perform a systematic review, as we wanted to be comprehensive in order to have the best possible overview of the knowledge available in the medical literature. The two aspects, systematic review and external validation, seemed sufficiently related to justify a joint manuscript.
  1. Wang, Yl.; Guilhaumou, R.; Blin, O.; Velly, L.; Marsot, A. External Evaluation of Population Pharmacokinetic Models for Continuous Administration of Meropenem in Critically Ill Adult Patients. Eur J Clin Pharmacol 2020, 76, 1281–1289, doi:10.1007/s00228-020-02922-z.
  2. Zhang, H.; Sheng, C.; Liu, L.; Luo, B.; Fu, Q.; Zhao, Q.; Li, J.; Liu, Y.; Deng, R.; Jiao, Z.; et al. Systematic External Evaluation of Published Population Pharmacokinetic Models of Mycophenolate Mofetil in Adult Kidney Transplant Recipients Co‐administered with Tacrolimus. Br J Clin Pharmacol 2019, 85, 746–761, doi:10.1111/bcp.13850.
  3. Mao, J.-J.; Jiao, Z.; Yun, H.-Y.; Zhao, C.-Y.; Chen, H.-C.; Qiu, X.-Y.; Zhong, M.-K. External Evaluation of Population Pharmacokinetic Models for Ciclosporin in Adult Renal Transplant Recipients. Br J Clin Pharmacol 2018, 84, 153–171, doi:10.1111/bcp.13431.
  • Our last literature query was at the end of 2020. This is a trade-off for the methodology we chose for this work. As the study progressed, we found that it was time-consuming to carry out the analysis with the necessary rigour. New work has been produced or is in progress since our last review request. However, to produce a new analysis based on a comprehensive review of the literature over this period would again take several months without ever being able to be definitively up to date. The following passage was added to the discussion of the manuscript to address this issue:

“Since our last literature search, new pharmacokinetic studies have been conducted or are underway. For example, a Pubmed/Medline search in February 2023 using the keywords "rivaroxaban population pharmacokinetics" for 2021-2023 identified 5 studies [45-49] that would have met our systematic review inclusion criteria by title and abstract. All were single-compartment models with first-order uptake and none were specifically designed for the obese patient population. The present study did not evaluate these models from studies conducted after our last literature search.”

The manuscript has been updated with the following bibliographic references :

  1. Zhang, D.; Chen, W.; Qin, W.; Du, W.; Wang, X.; Zuo, X.; Li, P. Population Pharmacokinetics and Hemorrhagic Risk Analysis of Rivaroxaban in Elderly Chinese Patients With Nonvalvular Atrial Fibrillation. The Journal of Clinical Pharmacology 2023, 63, 66–76, doi:10.1002/jcph.2145.
  2. Zhang, F.; Chen, X.; Wu, T.; Huang, N.; Li, L.; Yuan, D.; Xiang, J.; Wang, N.; Chen, W.; Zhang, J. Population Pharmacokinetics of Rivaroxaban in Chinese Patients with Non-Valvular Atrial Fibrillation: A Prospective Multicenter Study. Clin Pharmacokinet 2022, 61, 881–893, doi:10.1007/s40262-022-01108-3.
  3. Singkham, N.; Phrommintikul, A.; Pacharasupa, P.; Norasetthada, L.; Gunaparn, S.; Prasertwitayakij, N.; Wongcharoen, W.; Punyawudho, B. Population Pharmacokinetics and Dose Optimization Based on Renal Function of Rivaroxaban in Thai Patients with Non-Valvular Atrial Fibrillation. Pharmaceutics 2022, 14, 1744, doi:10.3390/pharmaceutics14081744.
  4. Liu, X.; Zhang, Y.; Ding, H.; Yan, M.; Jiao, Z.; Zhong, M.; Ma, C. Population Pharmacokinetic and Pharmacodynamic Analysis of Rivaroxaban in Chinese Patients with Non-Valvular Atrial Fibrillation. Acta Pharmacol Sin 2022, 43, 2723–2734, doi:10.1038/s41401-022-00892-9.
  5. Esmaeili, T.; Rezaee, M.; Abdar Esfahani, M.; Davoudian, A.; Omidfar, D.; Rezaee, S. Rivaroxaban Population Pharmacokinetic and Pharmacodynamic Modeling in Iranian Patients. Journal of Clinical Pharmacy and Therapeutics 2022, 47, 1284–1292, doi:10.1111/jcpt.13673.
  • The term PPK has now been defined in the abstract as population pharmacokinetics, we thank the reviewer for identifying this oversight on our part.
  • The observational data published in https://doi.org/10.1016/j.thromres.2021.10.009 have indeed been re-analysed. In order to provide more details, the following elements have been added as supplementary material to the manuscript:
    • The set of files used for the analyses performed with Monolix “monolix data.zip”
    • The R script used to calculate the covariates required by the different models (Cockcroft clearance, lean body mass, etc.) “dataman_R7.R”
  • A final paragraph has been added to clarify the message at the end of the paper, in line with the reviewer's recommendations:

“5. Conclusion

The study was not designed to draw clinical conclusions about the management of obese patients with VTE. However, the analysis provided information on the pharmacokinetics of the direct oral anticoagulants studied. Several population pharmacokinetic models of apixaban were applicable to the population of obese patients receiving curative apixaban treatment for VTE, suggesting that the results of these models developed in the general population are relevant to this specific population. In contrast, none of the models evaluated for rivaroxaban were applicable to obese patients treated for VTE. Extrapolations from these models, their parameter values or their simulation results should not be applied to obese patients treated with rivaroxaban for VTE.”